# Optimization of Polysaccharide Production from *Cordyceps militaris* by Solid-State Fermentation on Rice and Its Antioxidant Activities

**DOI:** 10.3390/foods8110590

**Published:** 2019-11-19

**Authors:** Ling Xu, Feng Wang, Zhicai Zhang, Norman Terry

**Affiliations:** 1School of Food and Biological Engineering, Jiangsu University, Zhenjiang 212013, China; lxu@ujs.edu.cn (L.X.); zhangzhicai@ujs.edu.cn (Z.Z.); 2Institute of Food Physical Processing, Jiangsu University, Zhenjiang 212013, China; 3Institute of Agro-production Processing Engineering, Jiangsu University, Zhenjiang 212013, China; 4Department of Plant and Microbial Biology, University of California, Berkeley, CA 94720, USA; nterry@berkeley.edu

**Keywords:** polysaccharide, solid-state fermentation, antioxidant activities, *Cordyceps militaris*, free radical

## Abstract

Polysaccharides are an important class of bioactive components of medical mushroom and herbs and are now used as natural drugs or dietary supplements on a global scale. In this paper, we aimed to increase the polysaccharide production of *Cordyceps militaris* and the antioxidant activities of fermented rice by solid-state fermentation. The media components and culture condition were optimized by orthogonal design and mono-factor tests using rice as the raw material. The optimal media consisted of (g/L): rice (50), fructose (7), glycerin (7), peptone (1), MgCl_2_ (0.11), VB_1_ (0.05), VB_2_ (0.05), CaCl_2_ (1.5), corn bran (6), and a water–materials ratio of 100%. The fermentation condition was as follows: inoculum volume of 5.5% (*v/w*), rice weight of 50 g in one bowl with a diameter of 120 mm and a depth of 90 mm, incubation temperature of 26 °C, and incubation time of seven days. Under the optimized condition, the maximal *C. militaris* polysaccharide content and free radical scavenging ratio were 68.3 mg/g dry substrate and 98.9%, respectively. This study provides a new strategy for the production of healthy food from traditional food.

## 1. Introduction

Medicinal mushrooms produce bioactive substances and have many medicinal functions such as lowering blood fat, fighting tumors, regulating immunity, and regulating metabolism [1,2,3,4]. Mushrooms have been used as drugs in ancient times [5,6]. Edible and medicinal mushrooms are an important part of natural drug resources in China and are a significant source of traditional Chinese medicine. Nowadays, medicinal mushrooms are also an attractive field for exploring new hypoglycemic drugs [7,8].

Medical mushroom polysaccharides are a class of biopolymers with high value and wide industrial applications [9,10]. The polysaccharides have many biomedical activities, including lowering blood sugar and antitumor and immunomodulating activities [11,12,13,14,15]. Mushroom polysaccharides can be used as drugs to treat diabetes, cancer, and other diseases. The polysaccharides can be extracted from their fruits, biomass [13], or broth in a submerged culture [16]. However, few studies have reported polysaccharide production by solid-state fermentation (SSF).

*Cordyceps militaris*, an insect pathogen of the Ascomycetes class, has been extensively studied. *C. militaris* has many pharmacological activities, including protecting the liver and kidneys and immunomodulation-related antitumor activities [17]. *C. militaris* has various bioactive constituents, such as cordycepin, polysaccharides, adenosine derivatives, ophicordin, and L-tryptophan. Polysaccharides of *C. militaris* (CPS) are one of the major bioactive components which possess antioxidant [18,19], immunomodulatory [12,16], antitumor [20,21], and anti-inflammatory [21] activities.

Rice is one of the major food crops and is rich in carbohydrates. China is the largest producer of rice, with over 30 million hectares of rice planting area, producing over 207.5 million tons annually. Rice contains a range of nutrients, including all amino acids, multiple B vitamins, and mineral elements. Thus, rice is extensively used in traditional Chinese foods. Traditional Chinese food (TCF), especially traditional fermented food, has played an important role in adding nutrients to the diet of people for centuries. Fermentation is an effective method used to produce bioactive peptides. Fermented TCFs, including fermented rice, soybean, meat, and milk, provide health benefits because of their bioactive components [2,7,22]. To date, no literature has reported on increasing the polysaccharide production of *C. militaris* by SSF in rice. The objective of the present study was to maximize CPS production and improve the antioxidant activities of fermented rice by the SSF of *C. militaris*. For this purpose, optimization of the culture media, incubation time, incubation temperature, and inoculum size were investigated.

## 2. Materials and Methods

### 2.1. Microorganism and Raw Materials

The strain of *C. militaris* (CGMCC 2909) was maintained by our laboratory. Rice was purchased from a local grocery store. We purchased 1,1-diphenyl-2-picrylhydrazyl (DPPH, CAS RN: 1898-66-4) and other chemical reagents from the Sinopharm Chemical Reagent Co. Ltd. (Shanghai, China).

### 2.2. Seed Culture

The slant seed culture was cultured on potato–dextrose–agar (PDA) slants at 26 °C for six days and then used as the seed of liquid seed culture. The seed culture medium was composed of (g/L): sucrose (20), peptone (5), yeast extract (5), wheat bran (10), KH_2_PO_4_ (2.4), and MgSO_4_ (1.2). The 250 mL Erlenmeyer flask was loaded with 100 mL of liquid seed medium. After autoclaving at 121 °C for 30 min, the seed medium was cooled to the room temperature and the slant seed culture was inoculated into the 250 mL flask. The flasks were incubated in the rotary shaker at 25 °C for five days and the culture was used as the inoculum for solid-state fermentation. The basal solid fermentation medium included rice (50 g/L), corn bran (4 g/L), and a water–materials ratio of 100%. In addition to rice and corn bran, various nutrients were added and mixed with water in inverted dome-shaped plastic bowls with a diameter of 120 mm and a depth of 90 mm. These plastic bowls were autoclaved at 121 °C for 60 min and were then sealed with fresh film. After being cooled to room temperature, 5 mL of seed culture was inoculated into solid fermentation media and the solid-state culture experiment was performed under the set condition. The cultured substrates were used for the analysis of CPS production and antioxidant activities. Three replications were used for all investigated factors.

### 2.3. The Orthogonal Experimental Design

The media components were key factors affecting CPS production and antioxidant activities of fermented substrates. According to the authors’ preliminary test, factors impacting the CPS production and antioxidant activities of fermented substrates included fructose (F_1_), glycerin (F_2_), peptone (F_3_), MgCl_2_ (F_4_), VB_1_ (F_5_), VB_2_ (F_6_), water–materials ratio (F_7_), CaCl_2_ (F_8_), and corn bran (F_9_). Therefore, a 4^9^-factorial design, or a factorial arrangement with nine factors at four levels, was employed to optimize the media components. The four levels of the nine variables are shown in the orthogonal designs (Table 1). A minimum orthogonal matrix method was selected as the L_32_(4^9^) (Table 2). Table 2 lists detailed experimental media components for each assay. The fermentation media of each run of orthogonal designs contained 50 g of rice and other defined components. All of the fermentation media were sterilized at 120 °C for 60 min. The fermentation was carried out at 25 °C for six days. The fermented substrates from each experiment were all subjected to polysaccharide extraction and starch hydrolysis, polysaccharide precipitation, and re-dissolution, CPS content test, and free radical scavenging ratio (FRSR) to DPPH determination.

### 2.4. Determination of CPS

In order to determine the CPS content of fermented substrate, the samples were dried at 80 °C to a constant weight. The dried samples were ground into a fine powder using a mortar and pestle. Subsequently, 20 mL deionized water and 0.1 g sample powder were mixed in a 100 mL Erlenmeyer flask, and the flask was heated in water bath at 80 °C for 6 h to extract the CPS. The CPS extracts were centrifuged at 5000 rpm for 10 min to remove the suspended solids. A total of 8 mL supernatant and 0.01 mL α-amylase (80,000 U/mL, Wuxi Enzyme Factory, Wuxi, China) were mixed and then kept at 80 °C in a water bath for 30 min. After being cooled to 60 °C, 0.01 mL glucoamylase was added to the mixture. The sample solution was continuously hydrolyzed for 60 min and then centrifuged at 5000 rpm for 10 min. Five milliliters of supernatant of the hydrolyzed sample solution was precipitated with the addition of 20 mL 95% ethanol, and then stored in the refrigerator at 4 °C for 24 h. After precipitation, the crude CPS was separated by centrifugation at 13,000 rpm for 15 min. The crude CPS were suspended in 1 mol/L NaOH in a bath at 60 °C for 1 h, and the solution was used to determine CPS concentration with the phenol–sulfuric acid method [23].

### 2.5. Antioxidant Activities Measurement

The FRSR of CPS toward 1-diphenyl-2-picryl-hydrazil (DPPH) was examined according to the method reported by Shimada et al. [24]. Two milliliters of 0.2 mmol/L DPPH ethanol solution was mixed with 2 mL of various concentrations of CPS solution at room temperature. After 30 min, the optical absorbance was measured at a wavelength of 517 nm. The low absorbance of the reaction mixture indicates higher FRSR. The scavenging of DPPH radical in percentage was calculated by the following equation: FRSR = (1 − A_1_/A_0_) × 100%, where A_0_ is the absorbance of the blank reaction and A_1_ is the absorbance in the presence of CPS.

### 2.6. Data Analysis

All tests were repeated three times and results were expressed as means ± standard error. All data were fitted using Excel software and SPSS software17.0 (SPSS Inc., Chicago, IL, USA).

## 3. Results and Discussion

An Erlenmeyer flask is usually used to optimize the culture condition and media components of fermentation. The main drawback of the Erlenmeyer flask is that its neck is narrow, which affects the aeration in the SSF. Thus, plastic bowls with a diameter of 120 mm and a depth of 90 mm were used to investigate the optimal SSF medium components and culture condition. Some parameters play dominant roles in the SSF process to increase the CPS production and antioxidant activities, including media component, inoculum volume, media loading amount, initial pH, incubation time, and incubation temperature. Initial pH or pH control in the fermentation process is the key factor to increase fungi polysaccharides [25,26] and antioxidant activity. However, a solid medium possesses a strong buffer capacity of pH. The adjustment of pH needs more acids or bases and has a negative effect on food flavor. In this study, the pH of the solid medium was set at a natural pH. Effects of media components, water–materials ratio, inoculum volume, media weight, incubation time, and incubation temperature on CPS and antioxidant abilities were tested.

### 3.1. Synergistic Effect of Components of Medium on CPS Production and Antioxidant Activities

Based on our previous tests, fructose, glycerin, peptone, MgCl_2_, VB_1_, VB_2_, CaCl_2_ had been identified as crucial factors affecting CPS production and antioxidant activities of fermented substrates. A low water–materials ratio directly affects the solubility, absorption, and metabolism of nutrients, while an excessively high water–materials ratio inhibits aeration and materials exchange, and further affects the fungi metabolism [19]. As a swelling agent, corn bran enhances fungi metabolism and improves the substrate utilization by increasing the aeration amount [27]. Therefore, a synergistic effect existed among water–materials ratio, corn bran, and other nutrient components of the media, especially carbon sources and nitrogen sources. Orthogonal design is an experimental method to study the synergistic effect of multiple factors at multiple levels. Its characteristics are high efficiency, speed, and economy. The design scheme and running results of L_32_ (4^9^) are shown in Table 2. The mean value of each level of every factor and the range value (*R* value) of each factor in L_32_ (4^9^) orthogonal design are listed in Table 3 to evaluate the effect of each factor on the two responses. The R value exhibits the importance of the factor’s action, and a larger R suggests the factor has a stronger effect on the CPS and antioxidant activities. It can be seen from Table 3 that the orders of all factors’ effects on polysaccharides and the FRSR were, peptone > CaCl_2_ > VB_1_ > VB_2_ > glycerin > corn bran > water–materials ratio > MgCl_2_ > fructose and glycerin > fructose > water–materials ratio > VB_1_ > CaCL_2_ > peptone > corn bran > VB_2_ > MgCl_2_, respectively. The two different orders suggested that other components with antioxidant activities existed in the fermented substrate. Peptone was the main factor that affected polysaccharide production. Lower peptone concentration was beneficial for the biomass growth but was adverse to CPS synthesis. Fructose and glycerin were the main factors to increase antioxidant activities. A higher concentration of fructose and glycerin enhanced antioxidant activities, which suggests that other components with antioxidant activities exist, and fructose and glycerin facilitate their synthesis. From Table 3, the optimal medium components for CPS production were (g/L): fructose 7, glycerin 1, peptone 1, MgCl_2_ 0.11,VB_1_ 0.05, VB_2_ 0.05, water–materials ratio 100%, CaCl_2_ 0.5, corn bran 4; and the optimal medium components for antioxidant activities were (g/L): fructose 7, glycerin 7, peptone 1, MgCl_2_ 0.08, VB_1_ 0.14, VB_2_ 0.05, water–materials ratio 100%, CaCl_2_ 1.5, corn bran 6.

Based on a comprehensive consideration of CPS production and antioxidant activities, the following medium was applied in further study about the effect of culture condition on the CPS production and antioxidant activities of fermented substrate (g/L): rice 50, fructose 7, glycerin 7, peptone 1, MgCl_2_ 0.11,VB_1_ 0.05, VB_2_ 0.05, CaCl_2_ 1.5, corn bran 6, water–materials ratio 100%.

### 3.2. Effect of the Inoculum Volume on CPS Production and Antioxidant Activities

Inoculum volume was also a significant factor in CPS production and antioxidant activities. Various inoculum volumes (4, 6, 8, 10, and 12 mL) were tested for their effects on CPS production and antioxidant activities (Figure 1). The higher CPS production (62.3 mg/g dry substrate) and FRSR (97.5%) were obtained at 8 mL of inoculum volume. It is important to provide an optimal inoculum volume in the fermentation process. A smaller inoculum volume may lead to insufficient biomass, causing the contamination of bacteria and other fungi, and reduce the product formation, whereas a larger inoculum volume results in too much biomass and the poor production of CPS [28].

### 3.3. Effect of the Rice Weight on CPS Production and Antioxidant Activities

In this study, we also focused on economic production. Therefore, the media loading amount is an important factor that effects production cost. Excessive media weight increases substrate thickness and inhibits aeration and heat diffusion, and finally decreases CPS production and antioxidant activities. For this purpose, we assessed the effect of media weight on CPS production and antioxidant activities of fermented substrate using rice weight instead of total media weight (Figure 2).

Various rice weight (30, 40, 50, 50, 70, and 80 g) were tested for their effects on CPS production and antioxidant activities (Figure 2). The higher CPS production (67.5 mg/g dry substrate) and FRSR (97.8%) were obtained at 60 and 50 g of rice, respectively. The result indicated that the media weight for the maximal CPS production was higher than that for the maximal FRSR, which suggested that the synthesis of CPS needs a relatively low aeration volume.

### 3.4. Effect of the Incubation Temperature on CPS Production and Antioxidant Activities

The maximal CPS production (67.21 mg/g dry substrate) and the maximum FRSR (97.27%) were attained at 26 °C (Figure 3). A reduction in CPS production or antioxidant activities was found when the incubation temperature was higher or lower than 26 °C. A higher temperature can be negative for the cells’ metabolic activities, and it has been reported that a lower temperature also decreases the cells’ metabolic activities. Hence, incubation temperature and its control in the SSF process was important for the CPS production and antioxidant activities. The poor heat diffusion in solid medium mean that the bio-heat derived from SSF process was accumulated in the medium, which leads to a reduction in microbial activity and decrease in CPS production and antioxidant activities [28,29].

### 3.5. Effect of the Incubation Time on CPS Production and Antioxidant Activities

High CPS production (67.86 mg/g dry substrate) and FRSR (97.52%) were reached after seven days of incubation (Figure 4). The incubation time is controlled by the fungi characteristics and is decided by the fungi growth rate and enzyme production. Some studies have reported that the fermentation time of bacteria or fungus was 48 h or eight to nine days, respectively [30,31].

## 4. Conclusions

The optimal medium for the maximal CPS production and antioxidant activities consisted of (g/L): rice 50, fructose 7, glycerin 7, peptone 1, MgCl_2_ 0.11, VB_1_ 0.05, VB_2_ 0.05, water–materials ratio 100%, CaCl_2_ 1.5, corn bran 6. The incubation condition was inoculum volume 5.5% (v/w), rice weight 50 g in a bowl of an inverted dome-shape with a diameter of 120 mm and a depth of 90 mm, incubation temperature 26 °C and incubation time of seven days. Under the optimal condition, the maximal CPS production and FRSR were achieved: 68.3mg/g of dry substrate and efficiency at the level of 98.9%. Therefore, the method for the production of fermented rice rich in CPS and antioxidant activity was developed by SSF with *C. militaris*. In addition to rice, there are a lot of stable foods with starch as main component, including corn, wheat, potatoes, and cassava. To increase the nutrients and functions of these stable foods, solid-state fermentation with these starchy foods by *C. militaris* is one suitable choice, and the optimized condition in this study can be used directly or as a reference. In addition, natural food fermented by edible and medical mushrooms under SSF can be applied in the production of nutritious food.

## Figures and Tables

**Figure 1 foods-08-00590-f001:**
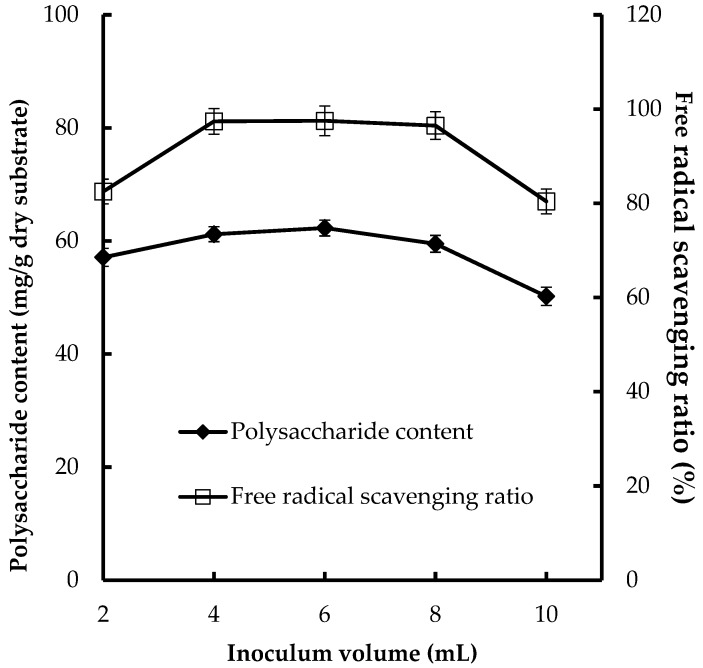
Effect of inoculum volume on CPS production and antioxidant activities of substrate.

**Figure 2 foods-08-00590-f002:**
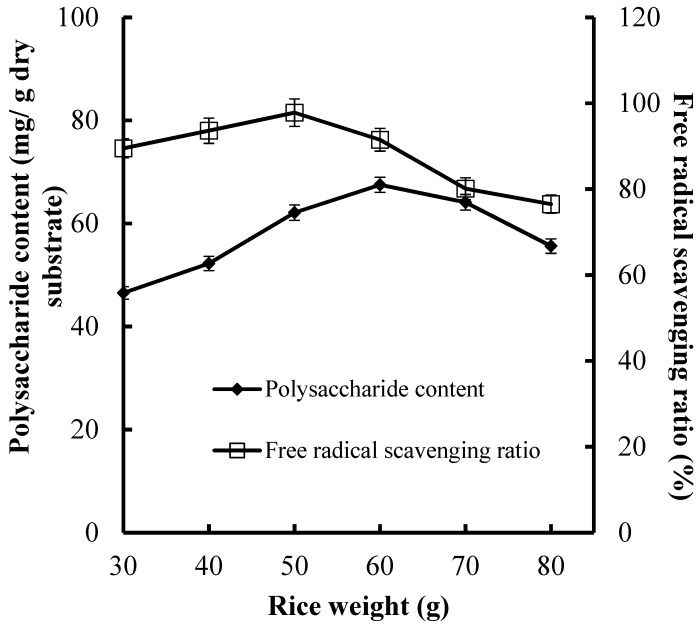
Effect of rice weight on CPS production and antioxidant activities of substrate.

**Figure 3 foods-08-00590-f003:**
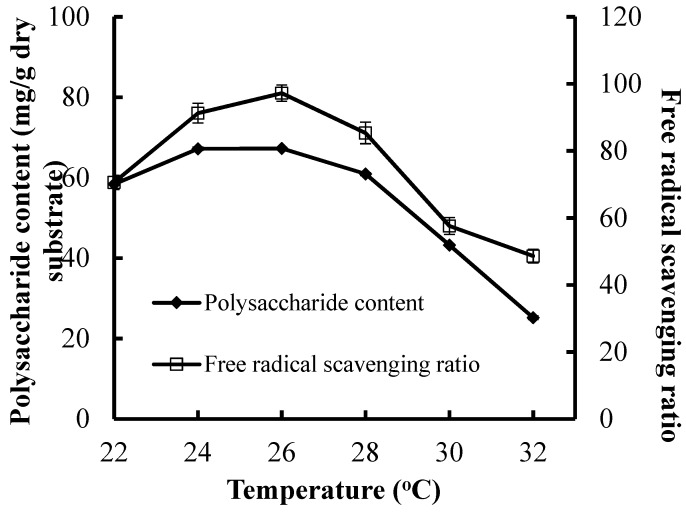
Effect of incubation temperature on CPS production and antioxidant activities of substrate.

**Figure 4 foods-08-00590-f004:**
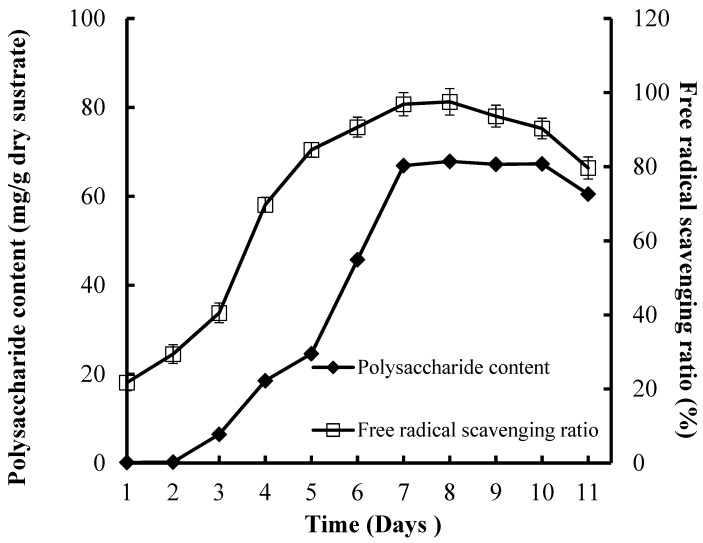
Effect of culture time on CPS production and antioxidant activities of substrate.

**Table 1 foods-08-00590-t001:** The factors and levels of the orthogonal design.

Factor	Number	Unit *	Level			
			1	2	3	4
Fructose	F_1_	g	1	3	5	7
Glycerin	F_2_	g	1	3	5	7
Peptone	F_3_	g	1	3	5	7
MgCl_2_	F_4_	g	0.05	0.08	0.11	0.14
VB_1_	F_5_	g	0.05	0.08	0.11	0.14
VB_2_	F_6_	g	0.05	0.08	0.11	0.14
Water–materials ratio ^a^	F_7_	%	80	90	100	110
CaCl_2_	F_8_	g	0.5	1	1.5	2
Corn bran	F_9_	g	3	4	5	6

* The rice amount of each run was set at 50 g. ^a^ Water–materials ratio was calculated according to total dry substrate weight.

**Table 2 foods-08-00590-t002:** Result of orthogonal design.

Run	F_1_	F_2_	F_3_	F_4_	F_5_	F_6_	F_7_	F_8_	F_9_	CPS ^a^	DPPH ^b^
1	1	1	1	0.05	0.05	0.05	80	0.5	3	58.465	82.0
2	1	3	3	0.08	0.08	0.08	90	1	4	41.787	86.1
3	1	5	5	0.11	0.11	0.11	100	1.5	5	23.508	91.4
4	1	7	7	0.14	0.14	0.14	110	2	6	17.375	94.8
5	3	1	1	0.08	0.08	0.11	100	2	6	24.817	90.6
6	3	3	3	0.05	0.05	0.14	110	1.5	5	28.720	92.9
7	3	5	5	0.14	0.14	0.05	80	1	4	25.229	93.8
8	3	7	7	0.11	0.11	0.08	90	0.5	3	25.811	96.8
9	5	1	3	0.11	0.14	0.05	90	1.5	6	36.065	96.1
10	5	3	1	0.14	0.11	0.08	80	2	5	26.902	95.0
11	5	5	7	0.05	0.08	0.11	110	0.5	4	22.853	97.1
12	5	7	5	0.08	0.05	0.14	100	1	3	23.411	97.3
13	7	1	3	0.14	0.11	0.11	110	1	3	21.399	67.6
14	7	3	1	0.11	0.14	0.14	100	0.5	4	44.696	90.9
15	7	5	7	0.08	0.05	0.05	90	2	5	20.017	92.3
16	7	7	5	0.05	0.08	0.08	80	1.5	6	24.987	96.0
17	1	1	7	0.05	0.14	0.08	100	1	5	27.484	57.2
18	1	3	5	0.08	0.11	0.05	110	0.5	6	22.732	53.1
19	1	5	3	0.11	0.08	0.14	80	2	3	27.581	58.4
20	1	7	1	0.14	0.05	0.11	90	1.5	4	40.065	71.2
21	3	1	7	0.08	0.11	0.14	80	1.5	4	33.156	65.3
22	3	3	5	0.05	0.14	0.11	90	2	3	23.265	68.8
23	3	5	3	0.14	0.05	0.08	100	0.5	6	36.647	78.2
24	3	7	1	0.11	0.08	0.05	110	1	5	36.041	75.3
25	5	1	5	0.11	0.05	0.08	110	2	4	32.356	64.0
26	5	3	7	0.14	0.08	0.05	100	1.5	3	44.938	83.2
27	5	5	1	0.05	0.11	0.14	90	1	6	23.872	80.0
28	5	7	3	0.08	0.14	0.11	80	0.5	5	28.526	90.4
29	7	1	5	0.14	0.08	0.14	90	0.5	5	37.835	77.1
30	7	3	7	0.11	0.05	0.11	80	1	6	36.987	93.5
31	7	5	1	0.08	0.14	0.08	110	1.5	3	38.999	94.7
32	7	7	3	0.05	0.11	0.05	100	2	4	36.938	95.1

^a^ The unit of polysaccharides of *Cordyceps militaris* (CPS) is mg/g dry substrate. ^b^ DPPH denotes free radical scavenging activity of CPS toward DPPH, and its unit is %. F_1_: fructose, F_2_: glycerin, F_3_: peptone, F_4_: MgCl_2_, F_5_: VB_1_, F_6_: VB_2_, F_7_: water content, F_8_: CaCl_2_, F_9_: corn bran.

**Table 3 foods-08-00590-t003:** Magnitude range analysis of orthogonal design.

	F_1_	F_2_	F_3_	F_4_	F_5_	F_6_	F_7_	F_8_	F_9_
Polysaccharides (mg/g dry substrate)									
k_1_	32.375	33.947	36.732	30.823	34.584	35.053	32.729	34.696	32.984
k_2_	29.211	33.753	32.208	29.181	32.605	31.872	31.090	29.526	34.635
k_3_	29.865	27.338	26.665	32.881	26.790	27.678	32.805	33.805	28.629
k_4_	32.732	29.144	28.578	31.299	30.205	29.581	27.559	26.156	27.935
*R*	3.521	6.609	10.067	3.7	7.794	7.375	5.246	8.54	6.7
DPPH (%)									
k_1_	74.3	75.0	85.0	83.6	83.9	83.9	84.3	83.2	81.1
k_2_	82.7	82.9	83.1	83.7	83.0	83.5	83.6	81.4	82.9
k_3_	87.9	85.7	80.2	83.3	80.5	83.8	85.5	86.4	84.0
k_4_	88.4	89.6	85.0	82.6	85.8	82.1	79.9	82.4	85.3
*R*	14.1	14.6	4.8	1.1	5.3	1.8	5.6	5	4.2

Note: F_1_: fructose, F_2_: glycerin, F_3_: peptone, F_4_: MgCl_2_, F_5_: VB_1_, F_6_: VB_2_, F_7_: water–materials ratio, F_8_: CaCl_2_, F_9_: corn bran; K1, K2, K3, and K4 were the average values of Level 1, Level 2, Level 3, and Level 4 for each factor.

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
