# Peer review of "Optimization of Polysaccharide Production from *Cordyceps militaris* by Solid-State Fermentation on Rice and Its Antioxidant Activities"

_foods, 2019, doi:10.3390/foods8110590_

Round 1

Reviewer 1 Report

This manuscript is on the media optimization for the production of polysaccharides.

There are really many papers on the optimization of media, but there is no novel findings in this manuscript.

Also, the validation of the result is not enough.

Figures 3 and 4 are not meaningful.

Author Response

We are grateful for the helpful suggestions from the reviewer. With respect to specific requests for clarification, the following revisions have been made.

Response to Reviewer 1 comments

Point 1: This manuscript is on the media optimization for the production of polysaccharides.

There are really many papers on the optimization of media, but there is no novel findings in this manuscript. Also, the validation of the result is not enough.

Response 1: Thanks for your kind suggestion. The aim of this study is to provide a new strategy for the healthy food production from traditional food. For this purpose, medium optimization method was used to obtain the best product. Since there are many published papers on the medium optimization for the production of bioactive component from medical mushroom, it indicated that medium optimization is still a good method to receive the optimal culture condition. To achieve the valid results in this research, the experiments were organized in logic, experimental methods were described in details and the data was analyzed and provided in tables and figures.

Point 2: Figures 3 and 4 are not meaningful.

Response 2: Thanks for your kind suggestion. As shown in Fig. 3&4, the production of polysaccharides and antioxidant activity were influenced by the culture temperature and culture period. Therefore, it is necessary to investigate these culture conditions and these results were useful for the control of culture temperature and period in future application.

Reviewer 2 Report

Dear Authors,

First of all I would like to congratulate you on an interesting article "Optimization of Polysaccharides Production from Cordyceps Militaris by Solid-State Fermentation on Rice and Its Anti-oxidant Activities".

However,  I have a few comments to your manuscript:

Methods

Line 126
In the sentence: All tests were repeated three times and... - "were" is missing?

    2. Results and discussion

line 134
You wrote: ... inoculum volume, media capacity, initial pH... 
What does it mean media capacity? Clarify it, please.

line 136
I think, that it should be : fungi polysaccharides (Plural) and antioxidant activity/capacity - Singular form?

line 144
... affects solubility ... not soluble

line 144
In the sentence:
Low water materials ratio directly affects the soluble, absorption and metabolism of nutrients, and excessive high water-materials ratio inhibits aeration and materials exchange, and further affects the fungi metabolism.
I would replace this and with whereas or while

line 146
You wrote that "As swelling agent, corn bran enhances fungi metabolism and improves the substrate utilization by increasing aeration amount." but you don't support this theory with the citation. Add the citation, please.

line 187
In the sentence: "So the media weight is an important factor that effects production cost".
Do you mean "media composition" or "media concentration" ?
... that affects?

line 202
I would replace the word "bad" with the "negative", which seems to be more scientific.

line 204
I suggest to combine the meaning in those two sentences: 
The poor heat diffusion in solid medium cause that the bio-heat derived from SSF process was accumulated in the medium. It leads the reduction of microbial activity and the decrease of the CPS production and antioxidant activities.

or if the mechanism described in the first sentence is well known, add the citation to support them.

3. Conclusions

In the second sentence of conclusions, there is a mixture of information concerning fermentation conditions and sizes of fermentation vessel. Analysing your data it is not possible to prove that the bowl with another size, like for example the diameter of 240 mm and a depth of 180 mm, will not be also optimal. Maybe you should recommend an optimal shape of bowl???
Try to rewrite also the last sentence, maybe this way:
"Under the optimal condition, the maximal CPS production and FRSR were achieved: 68.3mg/g of dry substrate and the efficiency on the level of 98.9%".

Best regards,

Reviewer

Author Response

We are grateful for the helpful suggestions from the reviewer. With respect to specific requests for clarification, the following revisions have been made.

Response to Reviewer 2 comments

In “Methods”

Point 1: Line 126, In the sentence: All tests were repeated three times and... - "were" is missing?

Response 1: Thanks for your kind suggestion. It has been revised.

In “Results and discussion”

Point 2: line 134, You wrote: ... inoculum volume, media capacity, initial pH... What does it mean media capacity? Clarify it, please.

Response 2: Thanks for your kind suggestion. “media capacity” has been revised to “media loading amount”.

Point 3: line 136, I think, that it should be : fungi polysaccharides (Plural) and antioxidant activity/capacity - Singular form?

Response 3: Thanks for your kind suggestion. It has been revised.

Point 4: line 144, ... affects solubility ... not soluble

Response 4: Thanks for your kind suggestion. It has been revised.

Point 5: line 144, In the sentence: Low water materials ratio directly affects the soluble, absorption and metabolism of nutrients, andexcessive high water-materials ratio inhibits aeration and materials exchange, and further affects the fungi metabolism.

I would replace this and with whereas or while

Response 5: Thanks for your kind suggestion. It has been revised.

Point 6: line 146, You wrote that "As swelling agent, corn bran enhances fungi metabolism and improves the substrate utilization by increasing aeration amount." but you don't support this theory with the citation. Add the citation, please.

Response 6: Thanks for your kind suggestion. The citation has been added and the following references were re-numbered.

Point 7: line 187, In the sentence: "So the media weight is an important factor that effects production cost". Do you mean "media composition" or "media concentration" ?

... that affects?

Response 7: Thanks for your kind suggestion. “media weight” has been revised to “media loading amount”.

Point 8: line 202, I would replace the word "bad" with the "negative", which seems to be more scientific.

Response 8: Thanks for your kind suggestion. It has been revised.

Point 9: line 204, I suggest to combine the meaning in those two sentences:

The poor heat diffusion in solid medium cause that the bio-heat derived from SSF process was accumulated in the medium. It leads the reduction of microbial activity and the decrease of the CPS production and antioxidant activities.

or if the mechanism described in the first sentence is well known, add the citation to support them.

Response 9: Thanks for your kind suggestion. Those two sentences have been combined in one sentence.

In “Conclusions”

Point 10: In the second sentence of conclusions, there is a mixture of information concerning fermentation conditions and sizes of fermentation vessel. Analysing your data it is not possible to prove that the bowl with another size, like for example the diameter of 240 mm and a depth of 180 mm, will not be also optimal. Maybe you should recommend an optimal shape of bowl?

Response 10: Thanks for your kind suggestion. It has been revised and an optimal shape of bowl was recommended.

Point 11: Try to rewrite also the last sentence, maybe this way:

"Under the optimal condition, the maximal CPS production and FRSR were achieved: 68.3mg/g of dry substrate and the efficiency on the level of 98.9%".

Response 11: Thanks for your kind suggestion. It has been revised.

Reviewer 3 Report

To demonstrate the importance of this study, the authors need to describe more fully how the optimization of polysaccharides and antioxidants here relate to other sources.  In Table 3, what does "k" stand for? 

Author Response

We are grateful for the helpful suggestions from the reviewer. With respect to specific requests for clarification, the following revisions have been made.

Response to Reviewer 3 comments

Point 1: To demonstrate the importance of this study, the authors need to describe more fully how the optimization of polysaccharides and antioxidants here relate to other sources. 

Response 1: Thanks for your kind suggestion. More discussion has been added in the section of “Conclusion”.

Point 2: In Table 3, what does "k" stand for? 

Response 2: Thanks for your kind suggestion. The notes for “k1-k4” have been added.
